# The Complex Diseases of *Staphylococcus pseudintermedius* in Canines: Where to Next?

**DOI:** 10.3390/vetsci8010011

**Published:** 2021-01-18

**Authors:** Stephanie A. Lynch, Karla J. Helbig

**Affiliations:** Department of Physiology, Anatomy and Microbiology, La Trobe University, Melbourne, VIC 3086, Australia; Stephanie.Lynch@latrobe.edu.au

**Keywords:** *Staphylococcus pseudintermedius*, methicillin resistance, antimicrobial resistance, canines

## Abstract

*Staphylococcus pseudintermedius* is a pathogenic bacterium of concern within the veterinary sector and is involved in numerous infections in canines, including topical infections such as canine pyoderma and otitis externa, as well as systemic infections within the urinary, respiratory and reproductive tract. The high prevalence of methicillin-resistant *Staphylococcus pseudintermedius* (MRSP) within such infections is a growing concern. Therefore, it is crucial to understand the involvement of *S. pseudintermedius* in canine disease pathology to gain better insight into novel treatment avenues. Here, we review the literature focused on *S. pseudintermedius* infection in multiple anatomic locations in dogs and the role of MRSP in treatment outcomes at these niches. Multiple novel treatment avenues for MRSP have been pioneered in recent years and these are discussed with a specific focus on vaccines and phage therapy as potential therapeutic options. Whilst both undertakings are in their infancy, phage therapy is versatile and has shown high success in both animal and human medical use. It is clear that further research is required to combat the growing problems associated with MRSP in canines.

## 1. Introduction

Over the last decade, *Staphylococcus pseudintermedius* has been identified as a bacterial species of concern within the veterinary sector. *S. pseudintermedius* is an opportunistic pathogen frequently isolated from healthy canines and, more importantly, associated with numerous infections in animals [1]. Dogs are the most common animal species infected with *S. pseudintermedius,* with 84.7% of all *S. pseudintermedius* isolates originating from canine diseases including skin, ear and urinary tract infections [2]. It has been reported that up to 97.8% of methicillin-resistant *S. pseudintermedius* (MRSP) isolates show multidrug resistance (MDR) to three or more antibiotics routinely used in veterinary medicine [2,3,4].

*S. pseudintermedius* was first isolated in 1976; however, it was formerly identified as *Staphylococcus intermedius* due to the morphological similarities between the two species [5]. In 2005, using a DNA–DNA hybridisation technique on *S. intermedius* isolates collected from animals, *S. pseudintermedius* was revealed as a novel species [6]. 

It is now known that *S. pseudintermedius* belongs to a collective known as the *Staphylococcus intermedius* group (SIG) which encompasses three distinct species, *S. pseudintermedius*, *S. intermedius* and *S. delphini* [7]. While all members within the SIG group have been shown to colonise numerous animal species, *S. pseudintermedius* is said to be the most common SIG species associated with animals—particularly, the most prevalent commensal bacterium in dogs [7,8,9,10,11]. Therefore, for the purpose of this review, all literature describing canine isolates formerly identified as *S. intermedius* will be referred to as *S. pseudintermedius*, unless otherwise shown by genomic investigation. This current review aims to discuss the various infections that *S. pseudintermedius* is associated with to corroborate the significant impact that the bacterium has on the veterinary sector, followed by a discussion on current and future treatment options against *S. pseudintermedius* infections in dogs.

## 2. *Staphylococcus pseudintermedius*: A Pathogenic Bacterium of Veterinary Concern 

Although *S. pseudintermedius* is primarily known for its pathogenic potential in canine infections, it is important to understand that *S. pseudintermedius* is also a significant member of the normal flora in canines [12,13,14]. Several studies have isolated *S. pseudintermedius* from 46–92% of healthy dogs, with the highest prevalence at the perineum (the skin between the anus and vulva/scrotum), followed by either the nasal or oral mucosa [12,13,14]. One study found that 0–4.5% of healthy dogs are colonised with MRSP as part of their normal flora [15]. Methicillin resistance in *Staphylococcal spp*. is known to alter the affinity to all β-lactam antibiotics. MRSP is a growing concern, with a recent study finding that 63% of *S. pseudintermedius* strains isolated from sick dogs were methicillin-resistant, with 78% of these isolates also described as MDR, being resistant to three or more antibiotic classes [16]. Additionally, MRSP can be transferred from sick dogs to otherwise healthy canines via direct transmission or indirect environmental transmission [17]. *S. pseudintermedius* transmission and subsequent colonisation may be associated with numerous infections, with skin infections being the most common (see Figure 1). However, *S. pseudintermedius* is also present as a pathogen in multiple other canine disease pathologies [17,18,19,20]. This highlights the fact that antibiotic resistance is a real concern moving forward in the veterinary space, especially for the treatment of *S. pseudintermedius* in canines, and adds perspective about whether antibiotics are a viable treatment option for the future of veterinary medicine. 

### 2.1. Canine Pyoderma

Canine pyoderma is one of the most common bacterial skin infections diagnosed in small veterinary medicine and is associated with redness, lesions, pain and inflammation [21]. Canine pyoderma can vary from moderate infections to severe infections and is triggered by underlying factors such as allergic skin disease, ectoparasites and endocrinopathies. This initiates the colonisation of pathogenic *S. pseudintermedius*, which is the most common pathogen associated with cutaneous infections, isolated as the predominant pathogen in up to 92% of canine pyoderma cases (see Figure 1) [22,23,24,25]. While it is evident that *S. pseudintermedius* is the predominant pathogen associated with canine pyoderma, it is also the most common commensal species in dogs, and there is currently a lack of clear evidence as to whether commensal species cause infection or if external isolates initiate infection. 

To assess this knowledge gap, multiple studies have compared the sequence diversity between commensal and pathogenic *S. pseudintermedius* isolates from the same canines; however, the collective evidence from these studies has not been able to conclusively answer this question. A handful of studies have shown no distinguishable differences between *S. pseudintermedius* isolates from healthy or atopic dogs using molecular techniques, therefore indicating that no specific strains or clusters of strains are associated with canine pyoderma [26,27,28]. More specifically, *S. pseudintermedius* isolates collected from the mucosa, a colonisation site for commensal species, and lesion sites from infected dogs in one study were either indistinguishable or closely related, perhaps indicating that the commensal *S. pseudintermedius* isolates may also be the causative agent in these pyoderma cases [26]. In contrast, isolates from canine pyoderma lesions have also been shown to be completely unrelated to mucosal isolates, suggesting either the commensal species mutate to become pathogenic or external isolates of *S. pseudintermedius* colonise to cause infection [26]. It is clear that longitudinal studies are required to truly assess genetic relatedness between commensal isolates and those involved in pyoderma cases, despite the fact that these studies will undoubtedly be difficult to perform given the inability to predict the onset of canine pyoderma, which would in turn require very large cohorts of animals to give meaningful study outcomes.

The rise in MRSP may explain the discordance between *S. pseudintermedius* species isolated from canine pyoderma. MRSP is isolated in up to 59% of canine pyoderma cases (see Figure 1), perhaps indicating that *S. pseudintermedius* isolates involved in infections may acquire genes required for methicillin resistance and appear unrelated to the commensal species [22,23,24,25,29]. Additionally, the acquisition of external strains that may lead to disease may be supported by two independent studies which have shown that MRSP from infected dogs can transmit to healthy contact dogs and the environment (e.g., sleeping and eating areas) [17,19]. In the majority of cases, the healthy dogs were only MRSP-positive when the infected dog was MRSP-positive, inferring contamination rather than colonisation in healthy dogs [17]. However, there was one case where the healthy dog remained MRSP-positive, resulting in an ear infection, even after the infected dog recovered [17]. This implies that *S. pseudintermedius* isolates recovered from canine infections that are unrelated to the dog’s commensal species may be caused by external species acquired from contact with infected dogs. However, this still leaves room for research into the factors that may contribute to the difference between acquired infection versus no infection upon contact of a healthy dog with an MRSP-infected dog showing disease pathology. However, these findings suggest that MRSP may have the ability to spread further within the dog community, which may affect the availability of treatment options to treat MRSP infections [17,19]. Additionally, MRSP isolates are often also multidrug-resistant, demonstrating high levels of antibiotic resistance against multiple antibiotic classes [30]. One study showed that the number and variety of antibacterial drug classes previously prescribed to dogs resulted in higher cases of MRSP, particularly those that received beta-lactam drugs and concurrent immunomodulatory therapy [30,31]. The evidence of high antibacterial resistance in pyoderma isolates is concerning from a treatment perspective, especially as *S. pseudintermedius* causes various other diseases throughout dogs.

### 2.2. Otitis Externa

Otitis externa (OE), or inflammation of the outer ear, is a disease routinely diagnosed in small veterinary practices [32,33,34,35]. Primary causes of OE are factors that initiate inflammation, including foreign bodies, such as grass awns, endocrinopathies including hypothyroidism as well as the presence of parasites [35,36,37]. However, the most frequent primary cause of OE highlighted across multiple studies is allergies, including adverse food reactions or atopic dermatitis [35,36,37], with studies showing that up to 75% of those diagnosed with OE were also diagnosed with atopic dermatitis [35,36,37,38]. Bacteria and yeasts, particularly *S. pseudintermedius* and *Malassezia spp*., respectively, two commensal species of the skin, are listed as secondary causes of otitis externa [34]. *S. pseudintermedius* is a predominant pathogen associated with OE, isolated from 20–94.3% of OE cases in canines (see Figure 1) [32,34,36,38,39,40,41,42,43]. The variation surrounding the prevalence of *S. pseudintermedius* in OE is still under investigation; however, geographical location has been suggested as a potential strong contributor, but with no influence on seasonal trend [40].

While dog breeds such as spaniels, German Shepherd and Shar-Pei are represented significantly more in OE cases [36,37,40], the breed and the age of the dog may also influence the type of pathogen present [36]. Interestingly, breed as a predisposing factor to OE may be explained by ear confirmation, particularly in spaniel breeds, with two independent studies identifying a significant increase in diagnosis frequency in dogs with pendulous ears, likely due to the moist, warm conditions facilitating secondary bacterial and fungal growth [37,40]. This is not surprising as the outer ear has a similar structure to the epidermis of the skin and, therefore, species that affect the skin, such as *S. pseudintermedius*, can also affect the external ear canal [35]. This is important for veterinarians to consider, as dogs with canine pyoderma are therefore at a higher risk of developing a secondary infection of *S. pseudintermedius* within the ear canal, which should be considered when prescribing treatment options for OE. Importantly, MRSP strains have been isolated in 10–48.1% of canine OE cases (see Figure 1) [2,39,41,42], with one study showing that all MRSP isolates were also multidrug-resistant, being resistant to two or more antibiotic classes [39]. This study also found that recent administration of beta-lactam antimicrobials significantly increased the frequency of methicillin and fluoroquinolone resistance [39], with the number and duration of prior exposures significantly increasing resistance to particular antimicrobial classes and the prevalence of methicillin resistance, respectively [39]. These studies highlight that antibiotic treatment of OE moving forward is likely to be plagued with difficulties and that novel treatment options to reduce antimicrobial resistance should be considered.

### 2.3. Urinary Tract Infections (UTI)

Urinary tract infections (UTIs), particularly those caused by bacteria, are another common diagnosis within small veterinary practices, with approximately 14% of dogs contracting a UTI within their lifetime [44,45]. Numerous bacteria species have been isolated previously from canine UTI cases, including *Enterococcus *spp., *Proteus* spp., *Staphylococcus *spp. and *Streptococcus *spp., with *Escherichia coli* identified as the most common uropathogen, isolated from up to 51% of canine UTIs [44,45,46,47,48,49]. More recently, *S. pseudintermedius* has been shown as the most common *Staphylococcal *spp. present in canine UTIs, with studies reporting a variable frequency of *S. pseudintermedius* isolation in 6.3–94.7% of UTIs in canines (see Figure 1) [45,46,48,49,50,51,52,53]. The large variation in *S. pseudintermedius* isolation has in part been explained in a recent multicentre study over a 6-year period across 14 European countries [53]. This study found that the bacterial species isolated from canine UTI cases varied based on geographical location, as did the antimicrobial resistance of the respective bacteria isolated [53]. Within this study, *S. pseudintermedius* was the most frequently isolated pathogen from UTI cases in most countries; however, this varied from 0% *S. pseudintermedius* isolation rate in Spain to as high as 94.7% in Italy; similarly, the isolation of MRSP varied from as low as 1.15% in Sweden to as high as 50% in Italy [53]. Within this study, the variation in antimicrobial resistance, particularly MRSP, was potentially attributed to countries, such as Sweden, following tighter regulations in regard to antimicrobial regulation and use, therefore resulting in lower resistance rates. However, differences in methods used to identify antimicrobial resistance were also reported across countries, which may have impacted the prevalence rates. In addition, the differences in methods used for urine sampling and bacterial isolation may have also affected the *S. pseudintermedius* prevalence rates [45]. This highlights the importance of unifying the methods of isolation and antimicrobial resistance characterisation for more accurate representations of *S. pseudintermedius* prevalence and resistance. Alarmingly, despite the variation in sampling and detection methods across the sector, multiple studies have reported high rates of MRSP and MDR *S. pseudintermedius* from UTIs in canines [45,46,52,54], with a significant increase in methicillin and gentamicin resistance in *S. pseudintermedius* isolates over a 16-year period, and a significant increase in fluoroquinolone resistance over the last 7-year period [45,46,52,54]. Additionally, there has been a temporal increase in MDR resistance in MRSP isolates, with all MRSP isolates in the study by Marques and colleagues displaying resistance to all antibiotics tested [52]. This is concerning as *S. pseudintermedius* has been isolated in up to 33% of recurrent UTI in canines [55], therefore alluding to the fact that antimicrobial resistance in *S. pseudintermedius* is impacting the resolution of UTI, resulting in major therapeutic limitations.

### 2.4. Respiratory Infections

Respiratory tract infections (RTI) in canines are relatively common and encompass various diseases including bacterial pneumonia, canine infectious respiratory disease complex (CIRDC) and viral infections; additionally, they are readily passed between dogs in social settings such as dog parks and boarding kennels [56]. There are numerous bacterial and viral pathogens that cause RTI in dogs, resulting in clinical symptoms such as coughing, sneezing or excess discharge [57]. Based on these symptoms, in addition to the medical history and physical examination of the patient, a presumptive diagnosis is generally made. However, to identify the causative agents or perform antibiotic susceptibility testing, samples of the airway lavage are generally used to culture the bacterial species [58,59]. As a result, studies have reported several bacterial species associated with RTI, particularly *Staphylococcus spp.,* including *S. pseudintermedius,* which is isolated in 9.3–60% of RTI in canines (see Figure 1) [50,56,57,58,59,60,61,62,63,64].

There are many factors influencing the prevalence of *S. pseudintermedius* in canine RTI, including the type of RTI diagnosed, which has been shown to affect the bacterial species isolated [58]. Interestingly, *Staphylococcus spp.* is more likely to be isolated from canines with aspiration pneumonia, usually caused by fluid from the stomach or mouth entering the lungs, compared to dogs that have community-acquired pneumonia [58]. The heightened prevalence of *S. pseudintermedius* in cases of aspiration pneumonia is likely due to the presence of *S. pseudintermedius* in the mouths of healthy canines, therefore entering the lungs and causing infection. This variation in dominant species present in different RTI was confirmed by a recent study which also found differences in bacterial communities between community-acquired pneumonia and secondary-bacterial pneumonia [59]. In particular, dogs with community-acquired pneumonia showed a loss of bacterial diversity and a dominant taxon [59]. Meanwhile, dogs with secondary-bacterial pneumonia also had a dominant species; however, usually those derived from the upper respiratory tract—for example, *S. pseudintermedius* from the mouth—thus indicating that bacterial symbiosis is a common phenomenon in canine bacterial pneumonia [59].

In addition to the type of RTI influencing the prevalence of *S. pseudintermedius* isolation, studies aiming to monitor the antimicrobial resistance patterns of canine isolates throughout Europe found a higher proportion of *S. pseudintermedius* isolates from RTI in Poland from 2008 to 2014, therefore suggesting that geographical location may contribute to variation in *S. pseudintermedius* isolation from canine RTI [56,60]. This variation due to geographical location was in agreement with a recent study which additionally found a potential association between RTI and the season or the age of the dog, albeit not statistically significant [62].

As mentioned, both bacterial and viral pathogens are associated with canine RTI; therefore, previous research has explored the potential interaction between bacterial and viral pathogens involved in respiratory infections. Preliminary results performed in a mouse model showed that mice co-infected with *S. pseudintermedius* and canine influenza virus (CIV) showed significant increases in bacterial and viral load in various organs compared to mice infected with *S. pseudintermedius* or CIV alone and, subsequently, a significant increase in lesion scores in the tissues of co-infected mice [61]. This indicates that the control or treatment of viral infections in addition to *S. pseudintermedius* infections in canine RTI are equally as important for a successful outcome [61]. However, this phenomenon would need to be looked at in dogs to see if these data are translatable to actual clinical data.

In regard to treatment, bacterial RTI in canines are currently prescribed antibiotics such as trimethoprim, amoxicillin-clavulanic acid or enrofloxacin, either as monotherapy or dual therapy, depending on the severity of the infection [64,65]. A recent study found that 99.4% of all isolates recovered from canine RTI were resistant to at least one antibiotic, with 64.7% of isolates listed as MDR, with *Staphylococcus spp.* making up 7.1% of the MDR isolates [62]. Importantly, there is a significant association between the sex of the dog or the geographical season and the presence of MDR isolates, which is important to take into account when prescribing treatment options [62]. This high rate of resistance was confirmed in a similar study, as an alarming 57.4% of dogs had a bacterial isolate that was resistant to the antibiotics that were previously or currently prescribed to that dog [58]. Therefore, confirming previous antibiotic administration can increase the resistance profiles of respiratory bacterial isolates [58]. With increasing trends of antibiotic resistance in respiratory isolates, studies suggest minimising this using broad-spectrum antibiotic use and to avoid using previously prescribed antibiotics, which may limit the treatment availability for bacterial RTI in canines in the future [56].

### 2.5. Reproductive Tract Infections

Previous research has identified an increase in the frequency of *S. pseudintermedius* in healthy dams isolated from vaginal samples, the placenta as well as colostrum and milk samples around the time of parturition [66,67,68,69,70]. However, the presence of *S. pseudintermedius* within the reproductive tract, such as the uterus and the mammary glands, has been associated with diseases in canines including pyometra and mastitis, respectively, which may result in complications including neonatal mortality [66,69,70,71,72,73]. Canine pyometra is an infection within the uterus of breeding female dogs, with *E. coli* and *Staphylococcus *spp., predominantly isolated from pyometra cases [74]. To date, two studies have isolated *S. pseudintermedius* in 10.5–18% of pyometra cases; however, these two studies had relatively small sample sizes and therefore may not accurately represent the prevalence of *S. pseudintermedius* in canine pyometra cases (see Figure 1) [73,75]. There have been no additional studies to explain the role of *S. pseudintermedius* in pyometra cases; therefore, further work is required to determine the pathogenicity of *S. pseudintermedius* in canine pyometra cases.

Additionally, *S. pseudintermedius* has also been isolated from canines with clinical mastitis; however, the prevalence of *S. pseudintermedius* in mastitis cases has not been well researched [66,69,70,71,72]. Despite the lack of research into prevalence, it has been shown that all dogs experimentally inoculated with *S. pseudintermedius* develop clinical mastitis, resulting in symptoms including painful, hot and inflamed mammary glands [72], thus indicating that *S. pseudintermedius* can be pathogenic in the mammary gland and may be responsible for many canine mastitis cases [72]. It has been shown that in limited cases, *S. pseudintermedius* is the causative agent of mastitis in cows; therefore, further research is required to determine how often *S. pseudintermedius* is present in canine mastitis and whether it is the causative agent in dogs also [76].

While the presence of *S. pseudintermedius* in the reproductive tract may cause infection in the female canines, it has also been shown that identical or closely related *S. pseudintermedius* strains have been isolated from the mother’s milk and vaginal tract and the puppies’ skin and placental samples, indicating that *S. pseudintermedius* may be transmitted by intrauterine or vertical transmission [67,77,78]. While, in many cases, such transmission results in the healthy colonisation of commensal *S. pseudintermedius*, in the puppies, it has been shown that the transmission of *S. pseudintermedius*, specifically MRSP, has been associated with premature death within the first 2–3 weeks of life, also known as neonatal mortality [66,79,80,81]. Multiple studies have reported outbreaks of neonatal mortality due to septicaemia, with *S. pseudintermedius* or MRSP isolated from the blood or organs of all deceased puppies [66,79,82]. Interestingly, it was found that *S. pseudintermedius* strains collected from the organs of puppies were found to be linked to isolates from the mother’s milk and vaginal samples, therefore indicating that the vertical transmission of pathogenic *S. pseudintermedius* can result in fatal sepsis in puppies [79,82]. However, the cause of neonatal mortality is multifactorial and, in addition to infections, congenital defects and low birth weight may also contribute to neonatal mortality [83]. Interestingly, puppies born with no detectable microbiota (including *S. pseudintermedius*) have a slower growth rate compared to those born with a microbiota within placental and meconium samples [67]. While this was not directly attributed to neonatal mortality, the lack of commensal species including *S. pseudintermedius* may be a contributing factor. Considering that the transmission of pathogenic *S. pseudintermedius* from the mother to her puppies may result in fatal sepsis, however, the lack of microbiota, including *S. pseudintermedius* may contribute to low birth weight and thus may lead to neonatal mortality. Therefore, there is a fine line between commensal *S. pseudintermedius* colonisation and pathogenic infection, with some studies suggesting that commensal colonisation of *S. pseudintermedius* may protect against pathogenic *S. pseudintermedius*, by bacterial interference [78]. Similarly to respiratory tract infections, it appears that symbiosis of commensal species is important and should be taken into consideration when developing treatment options.

**Figure 1 vetsci-08-00011-f001:**
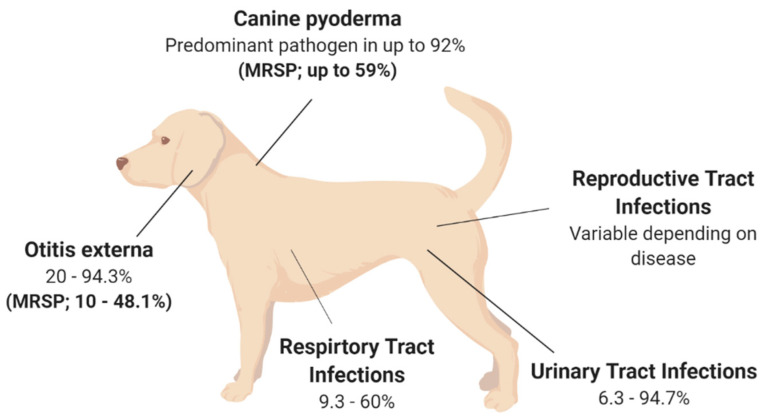
Isolation rates of *S. pseudintermedius* and MRSP from various disease states in canines. The isolation rates are presented as a range due to the variation described between studies; variation is described in detail in the main text. Canine pyoderma [22,23,24,25,29]; Otitis externa [2,32,34,36,38,39,40,41,42,43]. Urinary Tract Infections [45,46,48,49,50,51,52,53]; Respiratory Tract Infections [50,56,57,58,59,60,61,62,63,64]; Reproductive Tract Infections [66,69,70,71,72,73,76].

## 3. Current and Future Treatment Options for *Staphylococcus pseudintermedius* in Canines

*S. pseudintermedius* has the potential to cause or associate with a range of moderate to severe infections, of which, those left untreated, may have devastating outcomes. Therefore, future treatment options against such infections, especially in light of increasing antibiotic resistance, are a significant area of focus. It is important to note that the majority of studies on treatment options against *S. pseudintermedius* are in the context of canine pyoderma, as it is one of the primary reasons for antimicrobial prescription in the small animal veterinary sector [84]. Current guidelines created by diplomats of the American and European Colleges of Veterinary Dermatology state that canine pyoderma caused by *S. pseudintermedius* should be treated by topical and/or systemic antimicrobial therapy, as a gold standard [84]. A recent study found that 96.5% of dogs were prescribed antimicrobials for systemic and/or topical administration upon the diagnosis of canine pyoderma, with the majority of dogs receiving antibiotics including amoxicillin-clavulanate (55.7%), followed by cephalexin (43.9%) or clindamycin (10.0%) [85]. In addition to systemic antibiotics, 27.7% of canine pyoderma cases also received topical treatments, with active ingredients including fusidic acid, chlorhexidine and miconazole plus chlorhexidine shampoo [85]. Therefore, there is a significant need for alternative therapeutics. Two progressive areas of research for the treatment of canine pyoderma are vaccines and phage therapy, both of which have been successful in alternative animal diseases and offer promising alternative therapies for canine pyoderma and other *S. pseudintermedius* diseases moving forward. These novel therapies along with current therapies are discussed below.

### 3.1. Human Antibiotics

Antibiotics are the current frontline option against a multitude of infections in canines, due to their previously high efficacy, safety and ease of administration [86]. However, due to the rise in antibiotic resistance, particularly of *S. pseudintermedius*, against the majority of current antibiotic classes, research has focused on the potential veterinary use of new antibiotic classes or antibiotic classes that are currently reserved for human use. One of the new antibiotics, cefovecin, is a semisynthetic cephalosporin developed for use in cats and dogs and is effective in vitro against a broad range of Gram-positive and Gram-negative clinical strains obtained from Europe and the US [87]. Cefovecin has also been shown to have a long half-life, allowing for repeated doses in 14-day intervals, therefore reducing the frequency of antimicrobial exposure, which may reduce resistance rates [87]. When used in randomised clinical trials to treat dogs with superficial or deep pyoderma, cefovecin was shown to have a high clinical efficacy similar to previously used antibiotics, with only a small percentage of dogs showing adverse symptoms [88]. Although these results indicate that cefovecin is a safe and effective newly synthesised antimicrobial for veterinary use in particular geographical locations, the use of cefovecin is restricted only to use in cats and dogs where antibiotic sensitivity testing indicated use is required; therefore, additional research has focused on repurposing multiple human antibiotics for use in animals [89,90,91]. Many studies have tested the efficacy of various human antibiotics including nitrofurantoin, rifampicin, doxycycline, cefazolin and linezolid in the treatment of *S. pseudintermedius* infections in canines, mainly pyoderma, UTIs and surgical site infections [89,90,91]. Whilst studies did show that the use of such human antibiotics was effective in vitro against clinical isolates or was able to treat the infection of interest in clinical trials, it is important to note that some of these antibiotics resulted in adverse side effects and possessed a short half-life in serum, therefore requiring frequent and repeated doses, which is known to increase resistance [89,90,91]. Importantly, resistance was already noted against these tested antibiotics, either by the target strain, *S. pseudintermedius*, or associated species. For example, one study showed that rifampicin successfully treated pyoderma in 90% of dogs; however, in the 10% of dogs that failed to respond, *Proteus spp.* were isolated that were resistant to rifampicin [90]. The presence of resistance in the veterinary sector against human antibiotics is a major concern, as antibiotics such as nitrofurantoin, rifampicin, doxycycline and linezolid are essential in human medicine for the treatment of UTIs, tuberculosis in chemotherapy patients, chronic skin infections and pneumonia in humans, respectively; therefore, resistance would significantly impact the availability of treatment options in human medicine [89,90,91]. While alternative antibiotics may be effective in the treatment of *S. pseudintermedius* infections in dogs, particularly pyoderma, where other treatment options have failed, moving forward, the use of antibiotics intended for the treatment of human diseases may not be appropriate in veterinary medicine.

### 3.2. Topical Therapies

Recent guidelines state that topical therapy with proven activity against *Staphylococcus spp.* is the recommended treatment option against canine pyoderma and should be used in superficial cases of pyoderma where the owner can be compliant [92]. Currently, topical treatments are most likely used in combination with systemic antimicrobials, as topical treatments rapidly resolve lesions, show low rates of resistance and are shown to reduce the frequency and duration of antibiotics, which may decrease the evolution of antibiotic resistance [84]. Topical therapies are generally only sufficient for superficial canine pyoderma and include a multitude of formulations including shampoos, lotions, gels, creams and ointments with a range of antiseptic, disinfectant and active ingredients, as reviewed previously [93,94].

Shampoos are one of the most common topical therapies used for canine pyoderma as they are appropriate for large surface areas, act to remove both the bacteria and debris associated with the infection and are generally a cheaper option [94]. Previously, studies have shown that shampoos containing various active ingredients such as benzoyl peroxide, salicylic acid and chlorhexidine are effective in treating pyoderma by significantly improving bacterial counts and lesion scores, therefore successfully resolving canine pyoderma with no adverse side effects [95,96,97]. Importantly, combination treatment with a chlorhexidine shampoo and chlorhexidine digluconate spray also resolves pyoderma cases where MRSP is the causative agent and has been proven to be as effective as systemic antibiotic therapy [98]. One of the potential downfalls of shampoo formulas is the short contact time with the infected area before rinsing (~10 min); however, it has been shown that the fur collected from healthy dogs washed with the medicated shampoo shows antibacterial activity up to 17 days after the initial shampoo wash [99,100]. Therefore, in conjunction with previous results, the antibacterial activity of shampoo against pyoderma is safe, effective and long-lasting [99,100]. However, the administration of shampoos is generally only effective against superficial pyoderma and can be time-consuming, as throughout the studies, shampoos were generally applied multiple times a week for extended timeframes. This is concerning as the effectiveness of shampoos heavily relies on the compliance of the owners. Therefore, shampoos may not be appropriate as a sole treatment option for canine pyoderma, especially in more severe or chronic cases.

### 3.3. Vaccines

Vaccines are a common and effective therapeutic used in canines as a protective mechanism against an array of viral and bacterial diseases. The majority of licensed vaccines used in veterinary medicine are either live-attenuated or inactivated vaccines (bacterins) [101]. Such vaccines have been explored as a potential therapeutic for canine pyoderma.

#### 3.3.1. Bacterins

Bacterins are inactivated vaccines consisting of lysed suspensions of bacterial strains and adjuvants such as aluminium hydroxide, which aim to elicit an immune response within the host [101]. Research surrounding bacterin vaccines in animal therapy has been well explored. An early study demonstrated that *Staphylococcal* bacterin therapy, in addition to antibiotics and corticosteroids, promoted the regression of clinical signs of *Staphylococcal* blepharitis, with no adverse side effects observed [102], proving that bacterin vaccines may be a safe and effective option for further studies [102]. This paved the way for studies to assess the efficacy of bacterins in the form of vaccines to control recurrent pyoderma. Bacterin vaccines containing suspensions of *Staphylococcal* strains, in combination with systemic antimicrobial treatments, reduced clinical scores and demonstrated improved or complete remission for the majority of canines in multiple studies [103,104,105]. This treatment combination showed no adverse side effects. The commercialisation of a bacterin vaccine known as Staphage Lysate (SPL)^®^ has been a welcome addition to the new arsenal of treatment options for canine pyoderma. SPL is prepared by lysing cultures of *S. aureus* by *Staphylococcal* bacteriophages, followed by bacterial sterilisation [106]. One study found that SPL injections resulted in significant decreases in pruritus scores at weeks 12 and 23 of the treatment protocol [105]. Importantly, at the time of publication, no dogs from the study required follow-up antibiotics, indicating that the use of bacterins may prolong the recurrence of pyoderma and may reduce the frequency of antibiotics prescribed for recurrent pyoderma in the veterinary sector. However, this treatment may not offer a cure for recurring pyoderma infection caused by *S. pseudintermedius*.

A recent retrospective study analysed medical records from dogs that had received autogenous *Staphylococcus (pseud)intermedius* bacterin therapy for recurrent pyoderma that had not cleared in response to antibiotic therapy [107]. Bacterin therapy was administered subcutaneously to all dogs; however, the combined use of antimicrobials varied based on the veterinary surgeon’s discretion. Results from this study indicated that prior to bacterin therapy, 77.3% of dogs had a diagnosed or suspected allergic skin infection, with 64.7% of these dogs prescribed glucocorticoids or antibiotics concurrently with bacterin therapy for the management of the allergic skin disease. Throughout the 24-month study period, alongside systemic antibiotics, 68% of dogs required topical antimicrobial therapy, 36% of dogs required chlorhexidine-based shampoos, and a further 27% of dogs received antibacterial wash solutions. No adverse side effects were reported throughout the bacterin therapy protocol despite the variations in combination therapies used. Importantly, following bacterin therapy, 23% of dogs did not require follow-up antibiotics, with the remainder of the dogs prescribed significantly fewer courses and shorter exposure periods of systemic antibiotic therapy. However, 59.3% of dogs did require at least one repeat dose of bacterin therapy, with 50% of dogs requiring the repeat within 13 weeks of the first treatment [107]. These results are important, as although the findings indicate that less systemic antibiotics were required following bacterin therapy, there was still a reliance on topical and wash-based antimicrobial therapy to control canine pyoderma. Another crucial point discussed in this study was whether resistance genes, particularly in the context of MRSP strains, were destroyed during the production of bacterin therapeutics; therefore, further work is required to understand such factors [107].

It appears as though bacterin therapy is a safe and effective protocol in combination with alternative antimicrobial strategies for the management of *Staphylococcal* canine pyoderma; however, further development of this vaccine is required to release a product that will completely protect against pyoderma caused by *S. pseudintermedius*.

#### 3.3.2. Subunit

Subunit vaccines generally consist of specific part(s) of the target pathogen that are highly antigenic—for example, polysaccharides or proteins—to elicit an immune response within the host [101]. Vaccines against *Staphylococcal* infections have proved hard to develop and generally do not elicit a protective immune response against *Staphylococcal* infections [108]. One of the bacterial components that contributes to virulence and has been explored in both *S. aureus* and *S. pseudintermedius* is the presence of a specific cell wall anchored (CWA) protein, *Staphylococcal protein* A (spA) [109]. The spA in *S. aureus* and the orthologue spsQ in *S. pseudintermedius* is secreted during the log phase of bacterial growth and has been shown to have an immunosuppressive role [110,111]. This is because spA binds to the Fc domain of immunoglobulin (Ig) G to block opsonophagocytic killing and interacts with the Fab domain of IgM, resulting in B-cell superantigen activity, thus allowing the *Staphylococcal* spp. to evade the host immune system [110,111]. Previous research has administered an altered form of spA of *S. aureus* in both mice and guinea pigs [112,113]. Results show the production of antibodies that block the virulence of *S. aureus* and promote opsonophagocytic killing, therefore protecting the mice against challenge with a lethal concentration of methicillin-resistant *Staphylococcus aureus* (MRSA) [112,113]. As mentioned, *S. pseudintermedius* contains an orthologue of the spA protein, known as spsQ, and this cell wall protein is found in all clinical isolates of *S. pseudintermedius* [110]. Therefore, multiple research groups have explored the potential of protein A as a vaccine candidate for numerous *Staphylococcal* infections [110].

Currently, preliminary results show that canine IgG interacts with the cell wall protein of *S. pseudintermedius*, spsQ; however, this binding can be blocked by the presence of anti-protein A antibody, thus causing *S. pseudintermedius* to be more susceptible to phagocytosis [114]. Subsequently, when mutants of the spsQ were injected into clinically healthy dogs, there was a lowered toxic effect on canine B-cells, which resulted in a high titre of spsQ-specific antibodies which peaked 29 days after injection [115]. Therefore, the production of spsQ-specific antibodies may reduce immune suppression and establish a protective immunity against recurrent *S. pseudintermedius* infections. While results indicate that such vaccines against *S. pseudintermedius* are promising, it has been suggested that a *S. pseudintermedius* vaccine would likely contain additional virulence factor targets.

There is a lot of work still to be done on the subunit vaccine front in order to achieve a successful protective immune response against *S. pseudintermedius* infection in the canine. However, recent use of innovative approaches such as whole proteome and serological proteomic characterisation analysis on clinical isolates of *S. pseudintermedius* is yielding future potential targets for vaccine candidates.

### 3.4. Phage Therapy

Phage therapy is a therapeutic application of interest due to its success in both human and animal clinical trials. Phage therapy involves the use of small viruses called bacteriophages to kill specific strains of bacteria and has potential as an alternative treatment option for canine pyoderma [116].

Bacterio(phages) were first described by Fredrick Twort in 1915 as ultra-microscopic filter-passing viruses [117]. Phages were subsequently isolated by Félix d’Hérelle in 1917, and d’Herelle characterised the phages’ ability to cease bacterial cell development, resulting in lysis of the bacterial host (on an invisible microbe antagonistic) [118]. With over 100 years of research, it is now well understood that phages are highly abundant, small viruses that infect and replicate within their bacterial host and, in some instances, cause bacterial lysis, as reviewed Gordillo Altamirano and Barr [119]. Phage therapy for clinical infections focuses on lytic phages as their life cycle results in bacterial cell lysis [120], as shown in Figure 2, and will, therefore, be the focus throughout this section of the review.

From a clinical perspective, phages are favourable as they are highly host-specific. A recent in vitro study demonstrated that a phage cocktail was able to significantly reduce the levels of pathogenic *Escherichia coli* with no detectable impact on the six non-pathogenic *E. coli* species [121] compared to antibiotics that reduced all bacterial levels including commensals [121]. This indicates that the use of phage therapy is less likely to disrupt the host’s normal flora, thus reducing the likelihood of symbiosis or secondary infections, which is particularly important in the context of this review, as *S. pseudintermedius* is both a commensal bacterium and an opportunistic pathogen. In addition, phage therapy has been shown to be versatile and can be administered in the form of liquids, creams, aerosols, tablets and injections via numerous routes—for example, oral, subcutaneous and intravenous routes—depending on where the bacterial infection resides [122,123]. This is also important in the context of this review as infections caused by *S. pseudintermedius* occur in various regions of the body. Therefore, if the same phages could be used with altering formulations, this would significantly reduce barriers to the discovery and clinical development of new therapeutics for *S. pseudintermedius* canine infections.

There is significant research on phage therapy for human and animal infections with promising results; however, to the author’s knowledge, there has only been one veterinary clinical trial of phage administration in dogs to date. This preliminary clinical trial explored the efficacy of a phage cocktail containing six lytic phages against OE (otitis externa) caused by *Pseudomonas aeruginosa* [124]. Treatment using the phage cocktail resulted in a 30.1% reduction in clinical scores in the 10 dogs after 48 h [124]. Importantly, there was also a 67% reduction in *P. aeruginosa* counts after 48 h, and a parallel increase in phage titres, with up to a 100-fold increase in four dogs treated with the phage cocktail [124]. This phenomenon of increasing phage titres after administration at the infection site is known as auto-dosing and is another advantageous attribute of phage therapy.

Whilst there are limited studies of phage therapy in canines specifically, there has been an increase in clinical data for phage therapy in humans. A recent review shows the versatility and success of phage therapy in humans against numerous infections at multiple niches [125]. While human data on phage therapy cannot be directly translated to animals, these data show the promising potential of phage therapy for the multiple infections caused by *S. pseudintermedius* in canines.

Given the previously mentioned benefits and promising results of phage therapy in humans and animal clinical trials, it is not surprising that research has begun exploring bacteriophages as an alternative therapy for canine infections caused by *S. pseudintermedius*. While there are 19 phages that target *S. pseudintermedius* in the NCBI database, as displayed in the supplementary material by Zeman and colleagues, there are currently limited studies understanding these *S. pseudintermedius* phages [126]. The first known study characterising *S. pseudintermedius*-specific phages was published by Moodley and colleagues, who isolated four phages from canine faeces that were shown to preferentially lyse MRSP [127]. This was mentioned as a favourable attribute for phage therapy as these bacteriophages showed lowered lytic activity on methicillin-sensitive *Staphylococcus pseudintermedius* (MSSP); therefore, these phages would be less likely to lyse commensal MSSP strains.

Azam and colleagues also described four phages specific against *S. pseudintermedius,* of which three of the four phages (ɸSP120, ɸSP197 and ɸSP270) possessed a broad host range, as they showed strong infectivity activity against 95% of *S. pseudintermedius* strains, as well as moderate infectivity towards select coagulase-negative *Staphylococcal* (CoNS) species [128]. Whole-genome sequencing of these phages showed that there were no virulence, toxin or antibiotic-resistant genes present within the phage genomes, making them suitable candidates for phage therapy against *S. pseudintermedius* [128]. However, one downfall of all currently isolated phages against *S. pseudintermedius* is the fact that they are all temperate phages, containing integrase (CI) and repressor (Cro) genes within their genome. These genes indicate that the phages are able to undergo the lysogenic life cycle, which is unfavourable for phage therapy due to a lack of reliable bacterial killing. As mentioned by Moodley and colleagues, to remove the lysogenic life cycle potential of the phages, *vir* mutants can be selected via spontaneous mutation or by genetic modifications [127]. Random mutagenesis protocols for *S. pseudintermedius* temperate phage have been attempted, however, with no success to date [129]. Therefore, more research is required in optimising a protocol for site-specific modifications of the temperate phage genome to remove the lysogeny module or, alternatively, the isolation of truly lytic *S. pseudintermedius* phages.

To address the issues associated with the current *S. pseudintermedius* phages, two recent independent studies have explored the potential use of lytic *Staphylococcus aureus* phages with broad host ranges against *S. pseudintermedius* isolates [128,130]. Results from both studies showed that while the *S. aureus* phages tested (ɸSA039 and phiSA012) did possess a broad host range, able to infect *S. aureus, S. pseudintermedius, S. schleiferi* and CoNS, they both showed only weak to moderate infectivity towards *S. pseudintermedius* [128,130]. Therefore, further work is required to determine if the lytic activity of lytic *S. aureus* phages is sufficient enough to exploit their use in treating *S. pseudintermedius* canine infections. While phage therapy research has expanded, there are few clinical trials of phage therapy in companion animals, and as reviewed by Squires, there are numerous potential limitations that must be addressed to expand the use of phages as therapeutics [131]. In the context of canine pyoderma, it is important to understand the phage-resistant mechanisms and how this may impact the treatment of chronic or recurrent infections. Since it is known that pyoderma in dogs occurs due to underlying conditions, unless the underlying disease is well managed, the infection will recur, and it would be crucial to know whether the same phage formulation would be appropriate for re-treatment. Additional research is also required to assess the safety of using phages with a broad host range on the host flora and the potential infection of secondary opportunistic pathogens.

Interestingly, also within the realm of phage therapy, the use of endolysins as a novel antibacterial therapeutic has gained momentum, with studies assessing the potential of *Staphylococcal* endolysins against *S. pseudintermedius* canine pyoderma. As shown in Figure 2, endolysins are bacteriophage-encoded enzymes that are secreted and utilised by the mature phage at the end of the phage life cycle. Endolysins act to hydrolyse the peptidoglycan layer of the bacterial host from within, resulting in the destruction of the cell wall and the release of phage progeny, to infect neighbouring target cells and continue the phage life cycle [132]. One study expressed and purified the lysin (Lys-phiSA012) from the previously mentioned phiSA012 phage, which had somewhat weak activity against *S. pseudintermedius*, whereas its endolysins, Lys-phiSA012, had clear lytic activity toward the *S. pseudintermedius* isolates [130]. Similarly, a secondary study also expressed and purified endolysins from another *Staphylococcal* phage (*Staphylococcus* phage K); the endolysins showed lytic activity against the various *Staphylococci* strains obtained from canine pyoderma lesions [133]. Due to the efficacy of the endolysin in vitro, preliminary clinical trials were conducted on canine pyoderma lesions that had been present for almost a year and were resistant to all antibiotics tested [133]. The selected endolysin was formulated into a hydrogel and applied directly to the canine pyoderma lesions twice a day for 8 consecutive days. Results from the preliminary trial indicated that five out of the six dogs showed a 88–95% reduction in CFU of bacteria, with a subsequent reduction in redness and discharge from the lesion sites, and importantly, there were no allergic or adverse effects noted after 8 days of treatment [133]. This research highlights a novel contribution to an important issue of limited therapeutic availability; however, further work is necessary to characterise *S. pseudintermedius*-specific endolysins, to ensure that there are no off-target effects on the dogs’ commensal flora.

## 4. Conclusions

During this review, we have shown that while *S. pseudintermedius* have been isolated from healthy canines as part of the normal flora, *S. pseudintermedius* is also associated with a multitude of moderate to severe infections in dogs, particularly canine pyoderma. While studies support that *S. pseudintermedius* is required for a healthy microbiota in dogs, there is limited research to understand whether commensal *S. pseudintermedius* prevents or contributes to infections or, if commensal strains are involved in disease, which factors result in their pathogenicity. It is important to explore the involvement of commensal strains in disease progression as this would likely influence the type of therapy required to treat *S. pseudintermedius* infections. Whilst the exact mechanisms of infection are not well defined, it is now well known that *S. pseudintermedius* infections in canines are frequent and are commonly treated with antibiotics, and the misuse and overuse of antibiotics contributes to the global increase in antibiotic resistance, particularly the rise in MRSP. Additionally, *S. pseudintermedius* can transmit between dogs, the environment and to humans, thus showing that the presence of *S. pseudintermedius* is an ever-growing problem which impacts the availability of treatment options available in both the veterinary and human sector. Importantly, although *S. pseudintermedius* can transmit from infected canines to healthy canines, in the majority of cases, *S. pseudintermedius* colonisation does not result in disease in the healthy canines; therefore, further research into the factors that are associated with *S. pseudintermedius* infections is essential in understanding how to control and treat this pathogenic bacterium. Recent research endeavours have presented the field with some promising areas for future treatment options that will hopefully address canine infections caused by antibiotic-resistant *S. pseudintermedius* moving forward.

## Figures and Tables

**Figure 2 vetsci-08-00011-f002:**
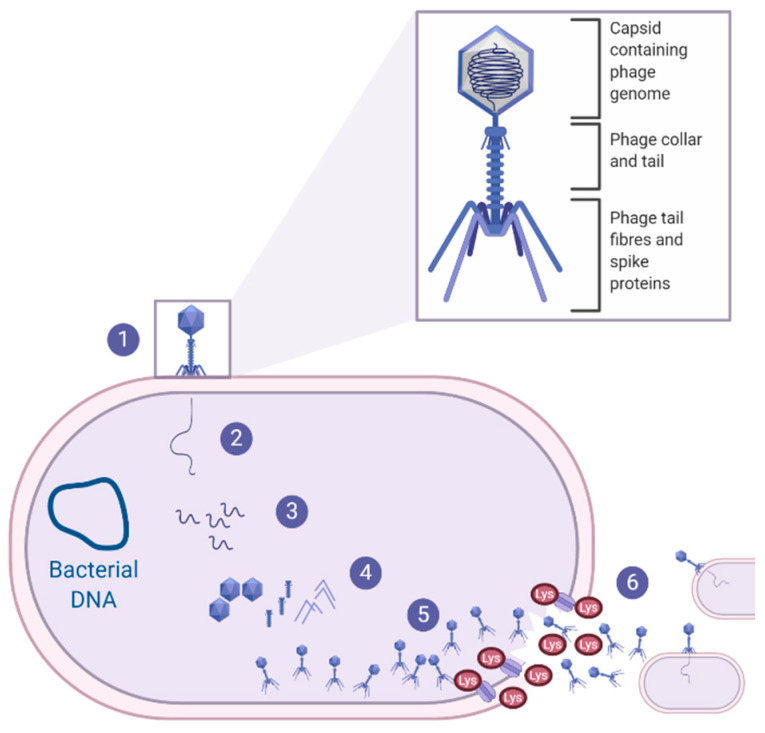
The lytic life cycle. Lytic bacteriophages undergo the lytic life cycle, which is initiated by (**1**) phage adsorption, in which the phage tail fibres recognise and attach to the bacterial outer membrane receptors. This results in an irreversible attachment of the phage to the bacterial surface which causes (**2**) the contraction of the phage tail sheaths to puncture the outer membrane and cell wall, allowing the phage genomic material to pass from the phage capsid to the bacterial cell via the phage tail. (**3**) The phage DNA is replicated using the bacterial host cell machinery and is subsequently (**4**) transcribed and translated into new viral components (e.g., capsid, tail, tail fibres). (**5**) Newly synthesised viral components will assemble to become mature virions which will secrete proteins known as holins and endolysins. (**6**) Holins will create a pore in the bacterial membrane, allowing the endolysins to pass through and degrade the peptidoglycan layer. This results in bacterial cell lysis, releasing hundreds of mature phage virions that will continue to infect and lyse neighbouring bacterial cells [119,120].

## Data Availability

Not applicable.

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
