# Peer review of "The Complex Diseases of Staphylococcus pseudintermedius in Canines: Where to Next?"

_vetsci, 2021, doi:10.3390/vetsci8010011_

Round 1
Reviewer 1 Report
The manuscript addresses an important topic and shed lights on a concerning rise of the problematic methicillin-resistant Staphylococcus pseudintermedius (MRSP) infections and the dearth of treatment options in vet. The authors clearly delineate this topic, therefore, this review is suitable for publication after addressing the below comments.
Comments:
-Line 59, define “MDR”.
-Lines 110-16; 305-7, lack of references.
-You have mentioned two different percentage ratios (Line 133; 147) for the S. pseudintermedius in OE, it should be similar.
-Line 158, replace “diagnosis” with “disease or infection”.
-Line 205, delete the parenthesis.
-Line 293, based on the context, the title should be rewritten to be “Current and future treatment options….”.
-Line 307, replace “are” with “is”. the administration of shampoos are generally…
-Line 509, I would assume “otitis” here is OE , please clarify this.
Author Response
Reviewer 1 :
We would like to thank the reviewer for their comments, and have addressed each point below:
- Line 59, define “MDR”.
This is defined on line 56
- Lines 110-16; 305-7, lack of references
These have now been added.
- You have mentioned two different percentage ratios (Line 133; 147) for the S. pseudintermedius in OE, it should be similar.
The two different percentages reflect isolation of S. pseudintermedius in OE, versus isolation of MRSP in OE, this explains the differences in the percentages in this instance.
- Line 158, replace “diagnosis” with “disease or infection”.
This sentence has now been removed as a result of comments by reviewer 2
- Line 205, delete the parenthesis.
This has been done
- Line 293, based on the context, the title should be rewritten to be “Current and future treatment options….”.  
This has been performed
- Line 307, replace “are” with “is”. the administration of shampoos are generally…
This has been done
- Line 509, I would assume “otitis” here is OE , please clarify this.
This has now been altered to read ‘OE (otis externa)’
Reviewer 2 Report
The manuscript covers the interesting topic of S. pseudintermedius infections in dogs. The issue has been well and comprehensively described, but I believe that the manuscript is too long and some sentences require little correction.
Authors should strictly stick to the subject and not describe widely side threads, e.g. about the vaccine based on S. aureus (P9, "The recent commercialization of a bacterin ... - reduce whole paragraph, please); the same remark applies to chapter 3.4. where the authors describe extensively the use of phage therapy in humans or its use to treat infections in dogs caused by bacteria other than S. pseudintermedius (Psedomonas aeruginosa). This is not a work on phage therapy, but about S. pseudintermedius - I suggest removing or reducing redundant information to a minimum. Some sentences are too long and their content may be unclear for the reader, eg P3: While in the majority of cases, the healthy dogs were only MRSP-positive .... "; "Additionally, MRSP isolates are often also multidrug resistant ....."; I suggest you refine the style of such long and slightly confusing sentences. P12, 3ed paragrafi "..... limited studies characterising ........ first known study characterising ....." (repeated "characterising", and characterising or characterizing?); P3- "The rise in MRSP may be one expalanation that could explain ..." - correct the style please.
What does the term "commensal sites" mean (P3, 2.1)? I think there is no such term in science.
P9-please provide a quotation on the methodology of preparing the Staphage Lysate vaccine - I am curious if bacterial antigens are not damaged during autoclaving?
Author Response
Reviewer 2:
We thank the review for their suggestions, and have addressed their points below:
- Authors should strictly stick to the subject and not describe widely side threads, e.g. about the vaccine based on S. aureus (P9, "The recent commercialization of a bacterin ... - reduce whole paragraph, please); the same remark applies to chapter 3.4. where the authors describe extensively the use of phage therapy in humans or its use to treat infections in dogs caused by bacteria other than S. pseudintermedius (Psedomonas aeruginosa). This is not a work on phage therapy, but about S. pseudintermedius - I suggest removing or reducing redundant information to a minimum.
We thank the reviewer for this suggestion, and this has now been completed.
- Some sentences are too long and their content may be unclear for the reader, eg P3: While in the majority of cases, the healthy dogs were only MRSP-positive .... "; "Additionally, MRSP isolates are often also multidrug resistant ....."; I suggest you refine the style of such long and slightly confusing sentences. P12, 3ed paragrafi "..... limited studies characterising ........ first known study characterising ....." (repeated "characterising", and characterising or characterizing?); P3- "The rise in MRSP may be one explanation that could explain ..." - correct the style please.
This has been performed here and throughout the manuscript where necessary.
- What does the term "commensal sites" mean (P3, 2.1)? I think there is no such term in science.
This term has now been removed.
- P9-please provide a quotation on the methodology of preparing the Staphage Lysate vaccine - I am curious if bacterial antigens are not damaged during autoclaving?
This has now been added, and reads: ‘SPL is prepared by lysing cultures of S. aureus by a Staphylococcal bacteriophages, followed by bacterial sterilisation [112]’